

# An open-source MEteoroLOgical observation time series DISaggregation Tool (MELODIST v0.1.0)

Kristian Förster[1,2], Florian Hanzer[1,2], Benjamin Winter[1,2], Thomas Marke[2], and Ulrich Strasser[2]

[1]alpS - Centre for Climate Change Adaptation, Grabenweg 68, A-6020 Innsbruck, Austria
[2]Institute of Geography, University of Innsbruck, Innrain 52f, A-6020 Innsbruck, Austria

*Correspondence to:* Kristian Förster (kristian.foerster@uibk.ac.at)

**Abstract.** Hourly meteorological time series are required in many applications in geoscientific modelling. These hourly time series generally cover shorter periods of time compared to daily meteorological time series. We present an open-source MEteoroLOgical observation time series DISaggregation Tool (*MELODIST*). This software package is written in Python and comprises simple methods to temporally downscale (disaggregate) daily meteorological time series to hourly data. *MELODIST* is
5 capable of disaggregating the most commonly used meteorological variables for geoscientific modelling including temperature, precipitation, humidity, wind speed, and shortwave radiation. In this way, disaggregation is performed independently for each variable considering a single site without spatial dependencies. The algorithms are validated against observed meteorological time series for five sites in different climates. Results indicate a good reconstruction of diurnal features at those sites. This makes the methodology interesting to users of models operating at hourly time steps who want to apply their models for longer
periods of time not covered by hourly observations.

## 1   Introduction

Continuous recordings of meteorological data are available since the late 18th century. During the 20th century, observational networks have been refined intensively, even at remote sites. However, these observations are generally not distributed equally in space and their temporal resolutions range from some hours (e.g., three measurements of temperature for each day) to one
15 day (e.g., rain gauges). Later, in the late 20th century, the instrumentation of meteorological stations has been supplemented by the installation of automatic weather stations (AWS) which are capable of collecting meteorological data continuously with a frequency ranging from one hour to one minute or even shorter periods of time (Rassmussen et al., 1993).

Figure 1 depicts the global temporal evolution of data availability for daily and hourly meteorological time series during the 20th century and beyond. This diagram has been compiled using two freely available datasets through querying the temporal
coverage of available data of each dataset: Daily data are collected continuously in the *Global Historical Climatology Network-Daily Database (GHCN)* (Menne et al., 2012; NOAA, 2015b), whereas the *Integrated Surface Database (ISD)* provides hourly time series of stations worldwide (NOAA, 2015a). This comparison reveals that the availability of hourly observations as provided by AWS is restricted to a few decades only. When observing Fig. 1, it becomes obvious that a large number of AWS have only been mounted in the last two or three decades.



Consequently, the question arises how to generate hourly time series of meteorological variables, e.g., by using available daily observations in order to benefit from their longer temporal coverage and higher spatial network density. In general, three completely different approaches exist:

1. Temporal disaggregation of daily meteorological observation (e.g., Waichler and Wigmosta, 2003; Schnorbus and Alila, 2004; Debele et al., 2007): This method is the simplest approach among the methods listed here even though more complex methodologies are also available, especially for precipitation (e.g., Koutsoyiannis et al., 2003). Simplicity holds, however, mostly true for computational needs as well as for the complexity of the methods itself. Deterministic equations or simple statistical models are applied to daily time series in order to derive hourly values. For each variable, the disaggregation is generally applied independently. Including statistical evaluations might improve results at a specific site compared to simple methods that are independent from station recordings (Waichler and Wigmosta, 2003).

2. Using weather generators to derive new synthetic time series that match the statistics of available hourly data: Weather generators calculate statistics of observed time series and apply these statistics using a random number generator to obtain new time series with equal statistical characteristics (Haberlandt et al., 2011; Ailliot et al., 2015). For hourly time steps, resampling techniques are applied in most cases (e.g., Sharif and Burn, 2007; Strasser, 2008). Time series derived by weather generators only match the observations statistically. The sequence of events is different due to its random nature. Weather generators are powerful tools that supplement deterministic modelling by stochastic methods and thus add a probabilistic component to the elsewise pure mechanistic methodology (mixed deterministic-stochastic models, see, e.g., Pechlivanidis et al., 2011). Combinations with disaggregation techniques are also possible (Mezghani and Hingray, 2009).

3. Dynamical downscaling using a local atmospheric model (LAM) of the atmosphere and atmospheric (re-)analysis data (e.g., Kunstmann and Stadler, 2005; Liu et al., 2011; Förster et al., 2014). As globally available data are used (e.g., re-analysis data), this approach is mostly independent of local observations although these local recordings might have contributed to the global datasets. It is a physically based approach that preserves physical consistency among all meteorological variables, which holds not necessarily true for the first and second methodology. However, due to its physical base, it is more complex and computationally expensive. Small scale precipitation might not be covered as accurate by the LAM in some cases due to the very complex micro-physical nature of precipitation and its variability (e.g., Förster et al., 2014).

In this study, we focus on the simplest method among the listed approaches, the disaggregation of daily meteorological data (# 1). For instance, in hydrological modelling, simple methods are usually sufficient in order to force conceptual, process-based models (Waichler and Wigmosta, 2003; Debele et al., 2007). To the authors' knowledge there is neither any "best" way of disaggregating meteorological data to hourly values nor any easy, ready to use and flexible software package that enables this task for different meteorological variables including precipitation, temperature, humidity, solar radiation, and wind speed. Therefore, we propose a robust and fully documented methodology including alternative approaches for all these variables in order to



make the best use of available data. Although there are more complex and sophisticated methods available for obtaining hourly values, *MELODIST* can be viewed as good balance among several aspects such as data availability, user's prior knowledge, robustness, and computational costs. Therefore, *MELODIST* addresses practitioners who need to run their model for long periods of time at one hour time steps. Here, emphasis is put on single stations rather than considering interdependencies among
different stations. However, the manuscript includes some specific remarks with respect to this restriction.

The paper is organised as follows: First, the study sites investigated herein are briefly presented in Section 2. The next section gives an overview of the disaggregation methods. In the fourth section, the methods are statistically evaluated with respect to their accuracy to reconstruct sub-daily features. Finally, Section 5 includes concluding remarks and an outlook for possible future work.

## 2   Study sites

The accuracy of disaggregation methodologies strongly depends on diurnal characteristics of meteorological variables. In turn, these diurnal characteristics might vary among different climates and environments. To test the robustness of the methods described in the next section, a small number of sites in different climates has been chosen (see, Fig. 2 and Tab. 1).

Except for Obergurgl, all station data are available for free. For each station, all relevant meteorological variables have
been recorded for at least one decade. Only shortwave radiation and precipitation are not available for Rio de Janeiro and Ny-Ålesund, respectively (Tab. 1).

The available datasets have been subdivided into two independent periods of time, one for calibration purposes, if required, and the other for an independent validation of the disaggregation results. This subdivision has been defined in order to enable a split-sample test (Klemeš, 1986) which requires an independent validation period for testing models. In this study, the split-
sample test is applied for the disaggregation methods described in the next section.

## 3   Disaggregation of daily to hourly meteorological values

### 3.1   Overview

In this section, all disaggregation methods employed in the framework of this paper are described in brief. For each meteorological variable different options are available (Table 2). Deterministic methods generally provide the same output if input remains
unchanged. In contrast, stochastic methods are based on random numbers. This means, that the output differs in consecutive runs even if the input dataset remains the same. Thus, stochastic methods require multiple runs prior to a sound statistical evaluation of these runs in order to draw conclusions. Some models require the calibration of model parameters that need to be adjusted for each site. Split-sample tests (Klemeš, 1986) are applied to test the methods more rigorously.



## 3.2 Temperature (T1)

Temperature on day $i$ is disaggregated to hourly values $j$ on using a cosine function whose amplitude is defined by the observed minimum $T_{\min,i}$ and maximum temperature $T_{\max,i}$ on day $i$ (e.g., Debele et al., 2007):

$$T_{i,j} = T_{\min,i} + \frac{T_{\min,i} + T_{\max,i}}{2} \cdot \left( 1 + \cos \left( \frac{\pi \cdot (t_j + a)}{12} \right) \right) \tag{1}$$

The parameter $a$ is determined either through providing an *a priori* guess of the temporal difference between the solar noon and the occurrence of the maximum temperature or through calibration. Three options are provided by *MELODIST*: Minimum and maximum temperatures occur at 7 am and 2 pm, respectively (T1a). The second option (T1b) relies on radiation geometry in order to calculate sunset as point in local time for minimum temperatures and sun noon + 2 hours as point in time for maximum temperatures (see, Fig 3). As the temporal shift of 2 hours might not be viewed acceptable as a general rule of thumb, temporal shifts for each month can be evaluated through statistical evaluation of observed hourly time series (T1c).

In principle, the methodology is based upon the assumption that the diurnal course of temperature simply tracks the diurnal course of the incoming shortwave radiative flux with a shift in time. This assumption does not hold true during polar nights which is why another method is applied for Ny-Ålesund. For this station, a linear interpolation between minimum and maximum temperature is applied (T1d nighttime option). If temperature increases compared to the previous day, minimum temperature is assumed to be representative for the first 12 hours of the current day and the maximum temperature is likewise attributed to the second half of that day. If temperature decreases from one day to the next, the opposite assignment is applied. Even though this method is rather simple, it preserves minimum and maximum temperatures while disaggregating.

## 3.3 Humidity

### 3.3.1 Humidity disaggregation based on dew point temperature (H1 to H3)

Relative humidity H [%] is defined as the ratio of actual vapour pressure $e_a$ [hPa] to saturated vapor pressure $e_s$ [hPa]:

$$H = 100 \cdot \frac{e_a}{e_s}. \tag{2}$$

It generally follows a diurnal course with the maximum around sunrise and the minimum in the early afternoon (Debele et al., 2007).

All humidity disaggregation methods require already disaggregated temperature recordings. Methods H1 to H3 generate hourly values of dew point temperature $T_{\text{dew}}$ [K], as the actual vapour pressure is assumed equal to the saturated vapour pressure at dew point temperature. Hourly $H$ values can thus be calculated using hourly values of $T$ and $T_{\text{dew}}$ as

$$H = 100 \cdot \frac{e_s(T_{\text{dew}})}{e_s(T)}. \tag{3}$$





Saturation vapour pressure for a given temperature $T$ [°C] is calculated using the Magnus formula (Alduchov and Eskridge, 1997):

$$e_s = \begin{cases} 6.1078 \exp\left(\dfrac{17.08085\,T}{234.175 + T}\right) & T \geq 0\,°\mathrm{C} \\[3mm] 6.1071 \exp\left(\dfrac{22.4429\,T}{272.44 + T}\right) & T < 0\,°\mathrm{C}, \end{cases} \tag{4}$$

while actual vapour pressure for a given temperature $T$ and relative humidity $H$ [%] is calculated as

$$e_a = e_s(T) \cdot \frac{H}{100}. \tag{5}$$

Methods H1 and H2 use a model in the form of $T_{\text{dew, day}} = a T_{\min} + b$ to calculate daily dew point temperature (i.e., no diurnal dew point temperature variation is assumed). For H1, $a = 1$ and $b = 0$, i.e., $T_{\text{dew, day}}$ is assumed to be equal to the daily minimum temperature. H2 uses hourly observations of temperature and humidity to calculate the best fit for $a$ and $b$ for a given site. $T_{\text{dew}}$ is thereby calculated from $T$ and $H$ by inverting eq. (4):

$$T_{\text{dew}} = \begin{cases} \dfrac{234.175 \ln\dfrac{e_a(T,H)}{6.1078}}{17.08085 - \ln\dfrac{e_a(T,H)}{6.1078}} & T \geq 0\,°\mathrm{C} \\[6mm] \dfrac{272.44 \ln\dfrac{e_a(T,H)}{6.1071}}{22.4429 - \ln\dfrac{e_a(T,H)}{6.1071}} & T < 0\,°\mathrm{C}. \end{cases} \tag{6}$$

H3 assumes a diurnal dew point temperature variation based on the assumptions that dew point temperature varies linearly between consecutive days, and that mean daily dew point temperature occurs around sunrise (Debele et al., 2007). Dew point temperature for a given day $d$ and hour $h$ is thereby calculated as

$$(T_{\text{dew}})_{d,h} = (T_{\text{dew, day}})_d + \frac{h}{24}\left((T_{\text{dew, day}})_{d+1} - (T_{\text{dew, day}})_d\right) + (T_{\text{dew},\Delta})_h, \tag{7}$$

where

$$(T_{\text{dew},\Delta})_h = \frac{1}{2}\sin\left((h+1)\frac{\pi}{k_r} - \frac{3\pi}{4}\right). \tag{8}$$

$k_r$ should be set to 6 for sites with average monthly radiation higher than $100\,\mathrm{W\,m^{-2}}$, and to 12 otherwise (Debele et al., 2007). An example application of these methods is shown in Fig. 4.

### 3.3.2 Minimum und maximum humidity disaggregation (H4)

Method H4 uses records of daily minimum and maximum temperature and daily minimum and maximum relative humidity as well as the disaggregated hourly temperature values to generate hourly humidity values:

$$H = H_{\max} + \frac{T - T_{\min}}{T_{\max} - T_{\min}}\left(H_{\min} - H_{\max}\right). \tag{9}$$

If $H_{\min}$ and $H_{\max}$ are available for each day, this method is the best available option among all available disaggregation methods (Waichler and Wigmosta, 2003).



### 3.4 Wind speed

Wind speed is a meteorological variable subjected to high variability at small temporal scales. This small-scale variability can be observed, e.g. from eddy-covariance measurements (Stull, 2009). The methods compiled in this study focus on suitable wind speed time series for hourly time steps without taking into account these small-scale considerations. This idea best corresponds to averages of wind speed (e.g., 10 minutes) rather single recordings carried out every hour, which might be mimicked by peaks related to small-scale variability or processes forced by larger scales.

#### 3.4.1 Equal distribution (W1)

As for precipitation, this method applies one unique value for each hour of the considered day. The daily mean value is assumed to be valid for hourly values as is (W1). For many applications, this assumption might be sufficient.

#### 3.4.2 Cosine function (W2)

Due to local and micro-climatic conditions, wind speed is subjected to diurnal variations on days with calm weather in absence of synoptic-scale weather patterns that overlie local and microclimatic forcings (Oke, 1987). Typical diurnal patterns in wind speed (and wind direction as well) are related to mountain-valley or land-sea wind systems. Besides these local climatic wind systems, wind speed typically increases during daytime and almost diminishes after sunset. This phenomenon is related to increased radiation-induced momentum flux on fair weather days. Again, synoptic scale weather patterns such as low pressure systems might overlie local-scale effects. These patterns of diurnal wind speed variations can be simply represented by a cosine function (W2), which requires calibration using data observed at the considered site. This model is similar to the temperature disaggregation method T1 (see, Eq. 1, Debele et al., 2007)

$$v_{i,t} = a_w \cdot v_i \cdot \cos\left(\frac{\pi \cdot (t - \Delta t_w)}{12}\right) + b_w \cdot v_i \qquad (10)$$

The wind speed representative for day $i$ is disaggregated to $v_{i,t}$ for hour $t$ (Fig. 5). $a_w$, $b_w$, and $\Delta t_w$ are parameters that need to be calibrated for each site prior to the application of this method.

#### 3.4.3 Random wind speed disaggregation (W3)

According to Debele et al. (2007) a random disaggregation of wind speed (W3) might also perform reasonably:

$$v_{i,t} = v_i \cdot [-\ln(\text{rnd}[0,1))]^{0.3} \qquad (11)$$

The function rnd is a random number generator which draws random numbers between 0 and 1. Figure 5 includes 10 runs (realisations) for this option. The daily average is not necessarily preserved by this method.





### 3.5 Shortwave radiation

#### 3.5.1 Radiation model and disaggregation of daily mean shortwave radiation (R1)

Shortwave radiation $R_0$ in $\mathrm{W\,m^{-2}}$ is computed for hourly time steps using the methodology described by Liston and Elder (2006), which predicts potential shortwave radiation $R_0$ for each time step. A simplified formula is provided that assumes a
flat surface (Liston and Elder, 2006):

$$R_0 = 1370\,\mathrm{W\,m^{-2}} \cdot \cos Z \cdot (\Psi_\mathrm{dir} + \Psi_\mathrm{dif}) \tag{12}$$

The solar constant ($1370\,\mathrm{W\,m^{-2}}$) is scaled according to the solar zenith angle $Z$, which depends on time (day of year and hour measured from local solar noon) and latitude (Liston and Elder, 2006). Details on these calculations as well as on the direct and diffuse radiation scaling values $\Psi_\mathrm{dir}$ and $\Psi_\mathrm{dif}$ are given by Liston and Elder (2006).
This methodology is applied for all three options. R1 assumes daily averages of shortwave radiation. This type of data is generally only available if hourly recordings of shortwave radiation have been aggregated prior to the data dissemination. In contrast, options R2 and R3 do not require shortwave radiation data as input.

#### 3.5.2 Disaggregation of sunshine duration (R2)

The method R2 builds upon the same methodology as R1 but runs the Ångström (1924) model prior to the disaggregation
computations. This model relates sunshine duration to mean shortwave radiation for daily time steps:

$$\frac{R}{R_0} = \left(a + b \cdot \frac{S}{S_0}\right) \tag{13}$$

Relative sunshine duration $S/S_0$ is transformed to relative global radiation $R/R_0$ and then the Liston and Elder (2006) radiation model is applied using this data. The parameters $a$ and $b$ are 0.25 and 0.75, respectively (Ångström, 1924). Figure 6 shows an example based on method R2 for summertime radiation in De Bilt (Fig. 2). The constants $a$ and $b$ have been obtained
through linear regression of $R$ and $S$ time series covered by the calibration period. If shortwave radiation and sunshine duration recordings are available, it is recommended to calculate these values for the site of interest.

#### 3.5.3 The Bristow-Campbell model (R3)

If radiation is not available, option R3 might provide reliable radiation estimates based on minimum and maximum temperature. It is assumed that small differences between maximum and minimum temperatures typically occur on cloudy days. However,
larger differences are common on sunny days with radiative cooling during nighttime and surface heating caused by shortwave radiative flux during daytime. The corresponding method is named after its inventors, Bristow and Campbell (1984):

$$\frac{R}{R_0} = A \cdot \left[1 - \exp(-B \cdot \Delta T^C)\right] \tag{14}$$





Here, relative global radiation $R/R_0$ is related to the diurnal temperature range $\Delta T$, which is estimated using maximum and minimum temperatures on specific day $i$ and the subsequent day $i+1$:

$$\Delta T_i = T_{\mathrm{max},i} - \frac{(T_{\mathrm{min},i} + T_{\mathrm{min},i+1})}{2} \tag{15}$$

Besides the parameters $A = 0.75$ and $C = 2.4$, which might be viewed as constants in a first step, $B$ is a site-specific parameter:

$$B = 0.0036 \cdot \exp(-0.154 \cdot \overline{\Delta T}) \tag{16}$$

In contrast to $\Delta T$, which refers to a certain day, $\overline{\Delta T}$ is the long-term average of differences between maximum and minimum temperature for the month of the current day. Based on these computations, radiation estimates are used as input to the radiation model R1 (see Fig. 6).

## 3.6 Precipitation

### 3.6.1 Equal redistribution (P1)

Reconstructing sub-daily precipitation intensities from daily values is challenging as precipitation intensities strongly vary in time and space. In the framework of this study, three methods are presented. The first method is the simplest way of disaggregating daily precipitation to hourly intensities by dividing the daily value by 24.

### 3.6.2 Cascade model (P2)

In order to provide a more sophisticated model that preserves sub-daily precipitation characteristics and is still less complex than typical weather generators, a simple statistical precipitation disaggregation approach has been set up: The cascade model by Olsson (1998). Some enhancements proposed in the literature (Güntner et al., 2001), such as weighting, have been taken into account as well. This method is a probabilistic approach providing different disaggregation results for each run (realisation). However, the statistical characteristics of each realisation are equal by definition.

The disaggregation is carried out assuming a doubling of temporal resolution for each step. Due to this stepwise doubling of resolution, the model is referred to as cascade model (see, Fig. 1 in Olsson, 1998). The time series of cascade level $i$ with time step $\Delta t_i$ is disaggregated to level $i+1$ with time step $\Delta t_{i+1} = \frac{1}{2} \cdot \Delta t_i$. The procedure is applied successively until the desired time step is reached. The doubling of elements of each subsequently derived time series implies that each box[1] of the higher level's time series has to be split in the next cascade level. Thus, the question arises how the separation of the precipitation volume $P_i$ into two temporally equally spaced time steps $P_{i+1,1} = w \cdot P_i$ and $P_{i+1,2} = (1-w) \cdot P_i$ (branching) is done, whereby

---

[1]The term box representing one data point, i.e. precipitation intensity for a given increment of time, is introduced by Olsson (1998) and, thus, herein used as well.





$w$ is the relative weight of branching for the first box of the subsequent level with respect to the total precipitation volume to be branched. Three cases are foreseen (Olsson, 1998):

$$w = \begin{cases} 0 & \text{with probability } P(0/1) \\ 1 & \text{with probability } P(1/0) \\ w_{x/x} & \text{with probability } P(x/x) \end{cases} \tag{17}$$

The first case indicates a branching that fills the second box of the subsequent level only, whereas the second case indicates the opposite. In contrast, the third case accounts for an weighted branching into both boxes of the subsequent level. For these cases, probabilities are provided for four different types of wet boxes with $P_i > 0$:

– starting box: This type of box indicates a dry box in the previous and a wet box in the next time step.

– ending box: An ending box follows a wet box and is followed by a dry box.

– isolated box: In this case, the adjacent boxes of the previous and the next time step are dry.

– enclosed box: The adjacent boxes of the previous and next time step are wet.

These probabilities for the three different branching possibilities (Eq. 17) can be achieved by an reverse scaling procedure. Highly resolved precipitation time series are aggregated by applying the cascade level branching assumption backwards. Every two boxes are added in each case representing the respective total volume of the antecedent higher level. Statistics are calculated for the branching types mentioned above (probabilities are derived through dividing counts of each case by the total number of elements of the time series). Separate evaluations are prepared for precipitation intensities below and above the mean precipitation value.

Additional statistics need to be computed for the case $P(x/x)$ for which the relative weight $w$ is evaluated as well. For all box types and both intensity classes, the relative weight ranging from zero to one is simply divided into seven bins (see, histograms in Olsson, 1998; Güntner et al., 2001) and counted according to the previously mentioned criteria (4 box types, 2 intensity classes, 7 classes of $w_{x/x}$). This procedure is applied for the aggregation steps $1 \to 2\,\text{h}\ (2^1\,\text{h})$, $2 \to 4\,\text{h}\ (2^2\,\text{h})$, $4 \to 8\,\text{h}\ (2^3\,\text{h})$, $8 \to 16\,\text{h}\ (2^4\,\text{h})$, and $16 \to 32\,\text{h}\ (2^5\,\text{h})$. According to Güntner et al. (2001), a count releated weight is assigned to the probabilities $P(0/1)$, $P(1/0)$, and $P(x/x)$ in each aggregation step prior to averaging the probabilities of all steps. The same procedure is applied to the weights. Finally, as a result, matrices of probabilities and weights are derived that represent the station's precipitation scaling. The parametrization is done by applying the empirical distributions of $P(0/1)$, $P(1/0)$, $P(x/x)$, and $w(x/x)$ to a random number generator (without fitting analytical distributions).

In turn, these matrices of probabilities and weights are used to disaggregate daily time series. The type of branching is determined by drawing random numbers for each branching step incorporating the probabilities $P(0/1)$, $P(1/0)$, and $P(x/x)$, which are evaluated cumulatively. If the random number is within the range of $P(x/x)$, a similar procedure is applied to determine the weight $w$ using another random number. In contrast to the aggregation procedure, disaggregation is applied





including the following steps (see Fig. 7): $24 \rightarrow 12\,\text{h} \rightarrow 6\,\text{h} \rightarrow 3\,\text{h} \rightarrow 1.5\,\text{h} \rightarrow 0.75\,\text{h}$ (Güntner et al., 2001). The time series with a 45 minutes time step is equally distributed to time series with a 15 minutes time step. These, in turn, are simply aggregated to obtain time series with a one hour time step.

For all disaggregation steps described above, the cascade model preserves mass which means that the precipitation total of the disaggregated time series is equal to the respective value of the original time series. Despite its simplicity with respect to model complexity and parameter estimation (Molnar and Burlando, 2005), cascade models have been already used successfully in different climates (Güntner et al., 2001). In contrast to more sophisticated models, the autocorrelation structure might not necessarily preserved (Koutsoyiannis, 2003).

*Remarks on spatial representativeness:* If this procedure is applied to more than one station, the sub-daily temporal distribution of precipitation is randomly derived for each station. These spatial patterns do not represent the actual spatial structure of the events at sub-daily time scales. For practical applications at the meso-scale, it is therefore suggested, to redistribute the sub-daily intensities for each station according to the cumulative relative sum of the station that is subjected to highest daily precipitation depth (Haberlandt and Radtke, 2014), which can be performed using the method described in the next paragraph. Areal peak intensities at sub-daily time steps might be overestimated due to this assumption. A more sophisticated but much more complex approach that has been developed recently (Müller and Haberlandt, 2014) takes spatial consistency explicitly into consideration.

### 3.6.3 Redistribution according to another station (P3)

Finally, a third method is supplied that addresses the generally higher network density of precipitation gauges compared to other meteorological variables. If a mixed network including hourly and daily observational sites is considered and if the distance among these stations is small, the relative mass curve of the station recordings at one hour time step can be transferred to the other sites for which only daily recordings are available. The values for the target sites are obtained through multiplying the relative mass of the highly resolved station's curve with the daily precipitation depth observed at the target site. This methodology is also applied in the tool IDWP, which is part of the hydrological modelling system WaSiM (Schulla, 2015).

## 4 Results and Discussion

### 4.1 Overview

This section follows the same structure as the methodology section. For each variable long-term averages of disaggregated and observed time series are presented and evaluated in order to assess the model skill of the disaggregation methods. Emphasis is put on prediction of diurnal features since most methods described herein are founded upon assumptions that imply a certain diurnal course for a given variable. This holds especially true for temperature, humidity, wind speed, and radiation. For precipitation, results are compiled and discussed for the cascade model. Due to the involvement of a random number generator in this method, evaluations with respect to model skill require the analysis of multiple runs (realisations).



Not all methods provided by *MELODIST* are evaluated. We focus on a subset of methods which might be relevant to a broad range of users with respect to typical data availability settings and typical applications. For each variable, the same methodology is applied to all stations listed in Section 2.

In order to put light on the model skill in a more quantitative way, statistical parameters have been derived for both the observed and the disaggregated time series (see, e.g., Tab. 3). All statistical parameters refer to the validation period listed for each station in Tab. 1 and have been calculated for hourly time steps. The mean value as well as the standard deviation have been computed for both time series for each station and each variable. The comparison of mean values gives an idea about possible biases, whereas the comparison of standard deviations is relevant to assess the comparability of the variability inherent in both time series. Moreover, the Root Mean Square Error (RMSE), the correlation coefficient $r$, and the Nash-Sutcliffe model Efficiency (NSE) have been calculated based on observed and disaggregated time series.

RMSE is a measure of deviations between observed and disaggregated time series on an hour-to-hour basis. Smaller values are generally better than larger values. The correlation coefficient is ideally close to one and describes the coincidence of phase for two series without considering biases. In contrast, NSE can be viewed as a combined measure addressing deviations in terms of biases and shifts in phase. It ranges from negative infinity indicating a low skill to one indicating a perfect fit. A value of zero means that the model is a as good as applying the average value.

## 4.2 Temperature

Despite the fact that only one option is available for temperature (T1), the standard-sine method enables different options to define the boundary conditions of the sine function (see Fig. 3). This method uses minimum and maximum temperature as input data. Here, results using the day length dependent option are presented, where maximum temperature is assumed to occur two hours after the solar noon. For Ny Ålesund, the modified nighttime option was activated as well in order to reliably disaggregate nighttime temperatures during polar nights, when the assumption of a distinct diurnal course does not hold true.

Long-term averages of hourly temperature derived for all sites are compiled in Fig. 8 alongside with the corresponding observations. The disaggregated diurnal course of temperature coincides well with observations for each station. Diurnal features are reliably preserved in the disaggregated time series. However, the amplitude is slightly overestimated for each site, attributable to the fixed assignment of minimum and maximum temperature for a given day of year. This assumption is mostly valid on fair weather days with surface heating but in some cases, e.g. when fronts cross the site of interest, minimum and maximum temperatures might occur at different times. Thus, minimum and maximum temperatures are more spread throughout the day in the observed datasets, resulting in a slightly smaller amplitude on average.

Besides this visual comparison, Tab. 3 summarises the model skill of temperature disaggregations for each station. Mean temperature values are well represented in the dataset given that the mean temperature was assumed to be unknown and only minimum and maximum temperatures have been involved in the analyses. The differences are smaller than 0.5 K. Due to the prescribed difference between minimum and maximum temperature, the standard deviations of observed and disaggregated time series are very similar. However, the magnitude of RMSE values shows that differences on hour-to-hour base exceed the



average bias. However, given that only two values per day are used as input data, the RMSE values can be viewed as good model performance. This holds also true for $r$ and NSE, indicating a high model skill.

Disaggregated time series of each station are of similar model performance. Only Rio de Janeiro has a slightly lower model skill, which can still be viewed as good model representation. Observations derived at Ny Ålesund indicate that even an

5 application of average values might be sufficient as disaggregation procedure, which can be explained by the lower impact of radiation on diurnal features of meteorological variables for that site. To conclude, temperature disaggregation based on minimum and maximum temperature should provide reliable estimates.

### 4.3 Humidity

As for temperature, Fig. 9 depicts the long-term mean of the diurnal course of relative humidity for all stations (H3 model).

The diurnal patterns of relative humidity are reasonably disaggregated through simulating a drop in humidity in the afternoon, which is observed at most stations. However, the accordance is less pronounced than for temperature. It is worth noting that the disaggregation of relative humidity depends on hourly temperature values. For these analyses, the results described for temperature in the previous sections have been applied for the disaggregation of relative humidity. Hence, uncertainties involved in the prior step also contribute to deviations between observation and disaggregation.

A closer look at the statistical evaluations derived for humidity disaggregation as compiled in Tab. 4 shows that the model performance is lower than the corresponding values obtained for temperature. The mean values are reproduced within a range of $\pm5\%$. Even though no information about daily minimum and maximum values of humidity have been involved in the disaggregation procedure, the standard deviations computed for observed and disaggregated time series are of similar magnitude. The RMSE amounts to 20% indicating comparably large differences between observed and disaggregated values even though

the mean bias is substantially lower. For all but one station, the correlation coefficient is higher than 0.5. In Ny Ålesund a correlation close to zero could be interpreted as inadequate model skill which is underlined when considering the negative NSE value. It may be assumed that the generally lower impact of radiation on other meteorological variables would suggest to use an equal redistribution of humidity values for that station.

However, the model performance achieved for the other stations is better given that the RMSE is lower and $r$ and NSE are

25 higher, respectively. In contrast to temperature, the humidity disaggregation performs best for Rio de Janeiro. To summarise, the disaggregation of humidity is reliable considering the fact that disaggregated temperature time series and only one humidity value per day have been used as input. These findings prove previous work that also discussed the accuracy of humidity disaggregation techniques (Waichler and Wigmosta, 2003; Bregaglio et al., 2010). If daily minimum and maximum values of relative humidity are available, the redistribution of these values should be pursued (see, Fig. 4 and Bregaglio et al., 2010).

### 30 4.4 Wind speed

Wind speed disaggregation has been accomplished using the modified sine-curve (W2). In Fig. 10 the long-term averages of the diurnal course of wind speed is plotted separately for observed and disaggregated wind speed, respectively. In this figure, wind speed is scaled as 'normative' wind speed, i.e. the value for each hour is divided by the mean value. Maximum wind



speed, which is typically observed during the afternoon hours, is well represented in the disaggregated time series. Small scale variability, as discussed in the methodology section, are not reproduceable by this approach.

As the mean value is simply redistributed according to a sine-function, mean values are exactly reproduced by the disaggregation approach. As already mentioned, variability (i.e. fluctuations) is neglected resulting in lower predicted standard deviations when compared to the corresponding standard deviations derived for the observed time series. If these fluctuations are not relevant for further evaluation, this disaggregation methodology for wind speed has an acceptable model skill which can be observed from the correlation coefficients and NSE values. Although these values are lower than those derived for temperature, they indicate a good model performance for all sites. The best model skill is achieved for De Bilt, whereas the lowest performance is achieved for Tucson, where a secondary wind speed maximum is observed in the morning. This diurnal pattern might be related to a local wind system that is subject to a change in wind direction and, hence, to a change in wind speed. Such phenomena are not addressed by this method.

### 4.5 Radiation

Even though radiation observations are available to most of the sites investigated in this study, the availability of daily mean shortwave radiation in absence of sub-daily time series is not so common. One exception is climate model output, which is typically aggregated to daily values. A typical real-world-case is, however, a long dataset of sunshine duration recordings. Therefore, method R2 is applied even though it is only applicable to De Bilt and Ny Ålesund. The diurnal course of mean hourly values derived through averaging the observed and disaggregated datasets is displayed in Fig. 11.

Given that the disaggregation is based on sunshine duration, the model skill can be viewed as very good for both sites. The timing of solar noon radiative fluxes as well as the phase of the disaggregated time series track observations very well which is also underlined by the performance measures presented in Tab. 6. Deviations between the mean values can be related to uncertainties involved in the Ångström (1924) model which has been fitted prior to disaggregation for both stations using the data from the calibration period. However, the disaggregated time series are subjected to similar variabilities as the observed time series which is expressed by the very similar standard deviations. As expected, the RMSE is comparably high when compared to the mean value of the time series since shortwave radiation is subjected to fluctuations due to the presence and absence of clouds causing rapid changes in shortwave radiation even for small increments in time. Notwithstanding these restrictions, the model skill expressed through the correlation coefficient and the NSE can be viewed as very good.

### 4.6 Precipitation

In contrast to the meteorological variables previously described, precipitation has been disaggregated using the cascade model (P2), which is a probabilistic model. As already explained, this change from deterministic to probabilistic methods requires a modified evaluation of model performance. Even though the precipitation total is preserved for each day throughout the disaggregation procedure, the occurrence and sequence of precipitation intensities differ from run to run. For rigorous testing and validation of the method, multiple runs are needed and their results have to be statistically evaluated. Here, the evaluation has been carried out according to the validation approaches described by Olsson (1998) and Güntner et al. (2001). Following




their ideas, Quantile-Quantile plots (Q-Q plots) of precipitation intensities are shown in Fig. 12, with close attention paid to the highest 1% of precipitation intensities. Since autocorrelation structure is not explicitly warranted by the cascade model, this feature is also tested (see, Fig. 13). As the common performance measures cannot be applied appropriately for random distributions of daily disaggregations, other performance criteria have to be considered. An approach similar to that described

by Olsson (1998) was chosen for that reason (see Tab. 7).

First, the simulation of peak intensities is studied through comparing observed and disaggregated intensities in a Q-Q plot (Fig. 12). For each station for which precipitation is available the highest 1% of disaggregated intensity values is plotted against the corresponding sorted time series of observed values. The cascade model was run 100 times, which is why 100 realisations are similarly evaluated. The areas shaded in light blue represent the range of values achieved through involving all realisations

in the analyses. In contrast, the area shaded in dark blue corresponds to the standard deviation of the considered quantile. Moreover, the mean of all realisations is drawn as black line for each station.

Even though intensity peaks are only represented implicitly through branching probabilities, precipitation peaks are well captured from a statistical point of view. For Rio de Janeiro, Tucson, and De Bilt, precipitation intensities are slightly underestimated. In contrast, an overestimation can be observed in the results of Obergurgl. The range of values indicate that some of

the highest values in the observed datasets are even exceeded in some realisations, which might underline the need for multiple runs.

Other characteristics that are also relevant for evaluations of sub-daily precipitation characteristics are summarised in Tab. 7. The mean duration of events ranges from 3 to 5 hours and is overestimated for all stations, which was also found by Olsson (1998) and Güntner et al. (2001). In contrast, the mean precipitation total of events derived through disaggregation is on average

similar to the respective observed value. This finding holds for all stations. It is evident that this value is higher in the subtropics than in the mid-latitudes. Although the total annual rainfall in Tucson is comparably small and the number of events per year is low, the average rainfall of events is also higher than in the mid-latitudes. This feature is correctly predicted by the cascade model. The duration of dry periods is also in good agreement compared to observations. Even though the length of events is over-predicted, the characteristics of the observed precipitation time series are captured very well for each site by the cascade

model.

To conclude, the cascade model preserves major characteristics of the observed hourly time series. However, these sub-daily characteristics can only be statistically evaluated due to the probabilistic nature of the approach. The model skill achieved for the stations listed in Tab. 7 can be viewed as reasonable reconstruction.

In addition, the autocorrelation structure is also validated (Fig. 13), as it is not explicitly preserved by the cascade model.

As for the intensity plot, shaded areas are added to the diagrams to show the variability in terms of total range and standard deviation of values, respectively. The autocorrelation derived for the disaggregated time series match observed values very well for Rio de Janeiro, Tucson, and De Bilt. For Obergurgl, higher $r_k$ values are observed which are not covered by the model results. The results derived using the cascade model for these sites can be viewed as good reconstruction of hourly precipitation features given that intensities, major characteristics of precipitation events, and the autocorrelation structure of

the disaggregated time series are in good agreement with observation.



## 5   Conclusions and outlook

The application of a simple but easy-to-use toolbox of disaggregation methods has been presented. Most of the methods included in *MELODIST* are parsimonious with respect to theory and computational costs. The basic levels of complexity have been chosen keeping practitioners in mind who need a package that is capable of disaggregating all relevant meteorological variables needed for environmental modelling. Available studies on disaggregation often focus on single variables such as precipitation rather than providing a unified framework for disaggregation. However, the presented package can be easily extended by more complex methods available in the literature as it provides basic functionalities for handling of time series with different temporal resolutions.

A set of methods relevant for real-world cases has been presented based on a split-sample test and statistical evaluations performed for the validation period. The presented methods perform well for different stations situated in different climates which underlines the robustness of the methods applied in the framework of this study. The highest model skill is achieved for temperature. Humidity disaggregation is, however, less reliable given that only one value per day is provided. The availability of minimum and maximum relative humidity improves the model skill. Wind speed disaggregation based on diurnal variations also works well if fluctuations are not required for further analyses. In contrast, the random wind speed function might be an alternative as it provides higher variabilities. Hourly radiation time series can be obtained with good agreement compared with observations, even if daily recordings of sunshine duration are used as input. Although precipitation was disaggregated using a stochastic approach which matches observations only in terms of long-term statistical evaluations, major characteristics of hourly precipitation features coincide well with observations. Based on this validation and the fact that different meteorological variables and stations have been involved in the validation analyses, *MELODIST* can be viewed as reliable and robust tool.

Some of the methods provided by *MELODIST* are based upon analyses of time series for parameter estimation, which requires a certain quality of data to derive sound parameters for performing the disaggregation runs. In general, it is important to note that data homogeneity might not always apply to long time series as changes in the instrumentation, micro-climate, and processing of data might have caused discontinuities in the time series (see, e.g., Rassmussen et al., 1993). For instance, Maturilli et al. (2013a) describe trends in the Ny Ålesund datasets, which are also tested herein. This is especially important if statistical disaggregation methods are applied that have been tuned for small periods of time only. Moreover, the limited availability of hourly observations involved in the statistics achieved in this study has to be carefully reviewed with respect to representativeness from a climatological point of view. In this study, different stations have been considered to investigate the robustness of methods rather than drawing conclusions in terms of climatic differences.

Homogeneity might be also relevant for disaggregation of time series that are subject to changes in climate. Ideas to cope with changing climatic conditions for disaggregation approaches are currently investigated. Two examples relevant in this context for the statistics based cascade model are a circulation-based parameterization in order to better predict changing weather patterns related to changing climate (Lisniak et al., 2013) and an intensity-based categorisation (Anis and Rode, 2014). Current research also focuses on the incorporation of the Clausius-Clapeyron relation to better predict rainfall intensities in future





climates (Bürger et al., 2014). These studies only address single stations or a limited study area without the consideration of different climates. Hence, the applicability of new methods should also be critically reviewed with respect to transferability.

In contrast to weather generators and dynamical downscaling approaches, physical consistency among the meteorological variables considered in this framework is not warranted. This limitation might restrict the methodology to derive input data only for conceptual models that are not pure physics-based approaches as the latter are more demanding with respect to this consistency. However, for most conceptual "grey box" models (see, e.g., Refsgaard, 1996) the quality of data provided by this disaggregation methods should be sufficient as tested in the framework of other model experiments (Waichler and Wigmosta, 2003). A better representation of the dependencies among the most relevant meteorological variables should be addressed explicitly in the future. Moreover, further emphasis should be on spatial consistency in disaggregation as already pursued by some authors (see, e.g., Koutsoyiannis, 2003; Müller and Haberlandt, 2014). The ongoing research on disaggregation methods underlines the need for sound and robust tools for disaggregating meteorological variables.

Even though *MELODIST* provides robust methods that do not include those very recent developments, it might serve as tool for both practitioners and scientists. For the latter group, *MELODIST* could be viewed as framework for performing future research on disaggregation since new disaggregations methods can be easily plugged in.

## Code availability

*MELODIST* is free open-source software and is licensed under the GNU General Public License version 3 (GPL3). The software package is written in Python and has been tested under Python 2.7 and 3.5. The packages `pandas`, `numpy`, and `scipy` are required as dependencies. How to get *MELODIST*:

- doi:10.5281/zenodo.46759

- https://github.com/kristianfoerster/melodist.git

- run `pip install melodist` on the command line

*Acknowledgements.* This work was carried out within the framework of the projects "W01 MUSICALS II - Multiscale Snow/Ice Melt Discharge Simulation for Alpine Reservoirs" and "W03 InsuRe II - Insurance Risk Evaluation of Flooding and Adaptation", carried out in the COMET research programme of the alpS - Centre for Climate Change Adaptation in Innsbruck, Austria. The authors want to thank the COMET research programme of the Austrian Research Promotion Agency (FFG) and the company partners TIWAG - Tiroler Wasserkraft AG and VLV - Vorarlberger Landes-Versicherung VaG. We also acknowledge the work of all the institutions that collect meteorological data and share these data with the public.



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





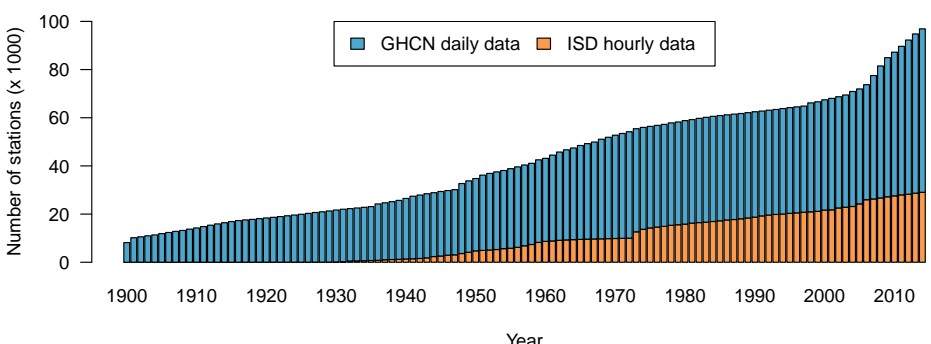

**Figure 1.** Time series of the worldwide station data availability in the 20th and 21st century according to the global ISD and the GHCN datasets.

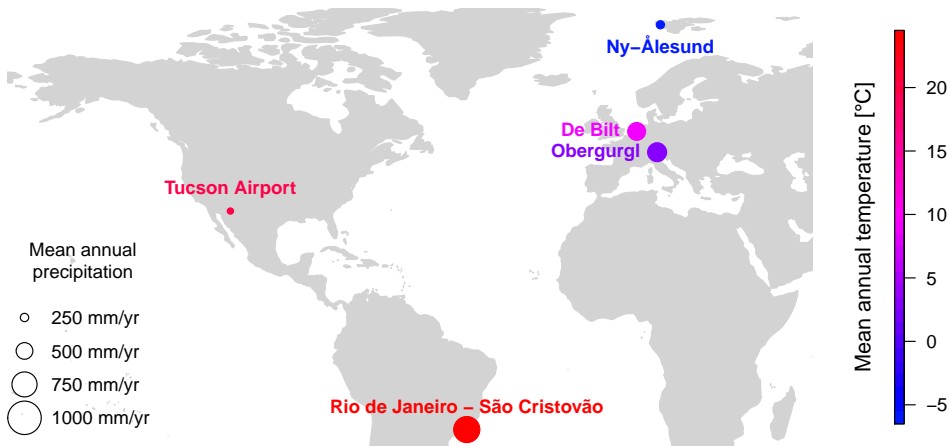

**Figure 2.** Map of stations investigated in this study. Dot size represents the mean annual precipitation, whereas the color of each station indicates the mean annual temperature.





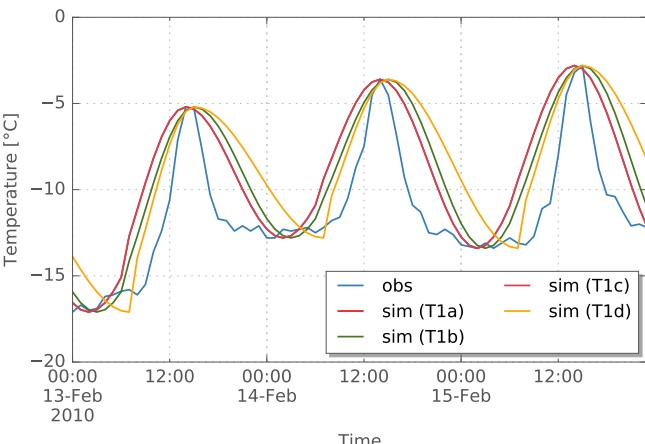

**Figure 3.** Example application of the temperature disaggregation model T1 performed with different settings for Obergurgl. Observed and disaggregated time series are shown for temperature. T1a: fixed abscissa values for minimum and maximum temperature; T1b: minimum and maximum temperature are related to sunrise and sun noon + 2 h, respectively; T1c: similar approach as T1b with an additional empirical shift of the maximum temperature; T1d: option T1a with modified nighttime option.

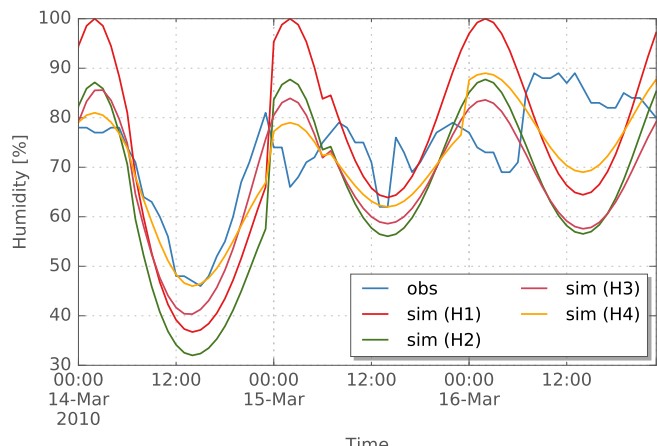

**Figure 4.** Example application of different humidity disaggregation models for Obergurgl. Observed and disaggregated time series are shown for relative humidity.





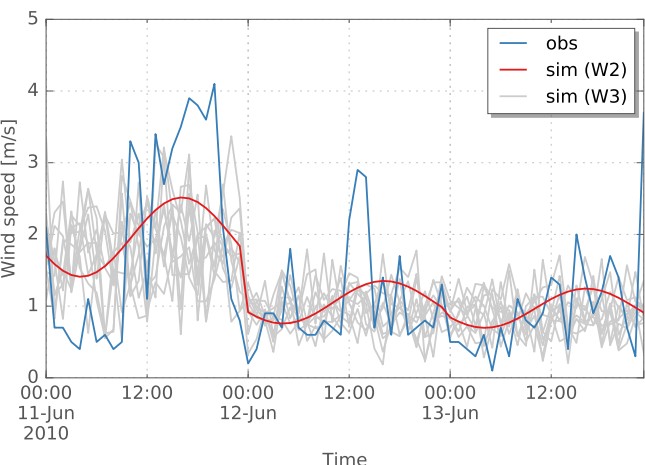

**Figure 5.** Example application of the wind disaggregation models W2 and W3 for Obergurgl. Observed and disaggregated time series are shown for wind speed. For option W3, 10 realisations are shown.

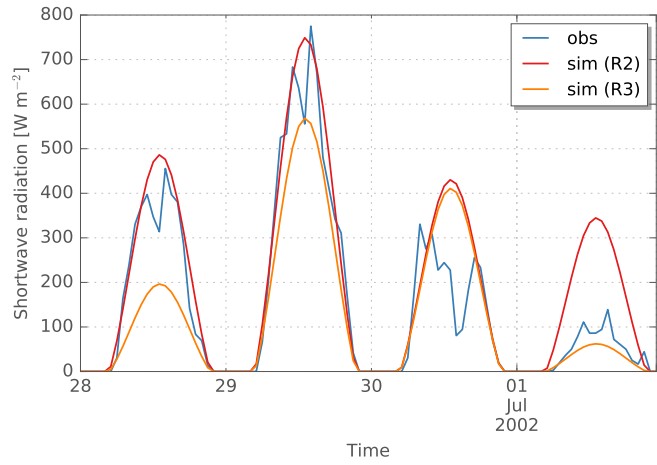

**Figure 6.** Example application of the radiation disaggregation model R2 and R3 for De Bilt. Observed and disaggregated time series are shown for shortwave radiation. Option R2 is based on sunshine duration, whereas option R3 requires minimum and maximum temperature as input.





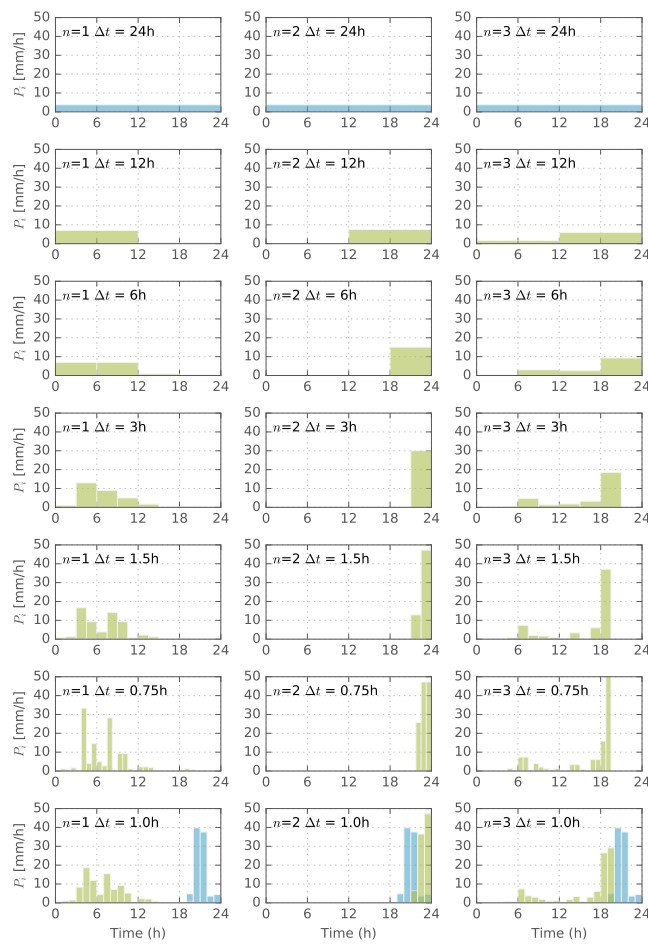

**Figure 7.** Example for precipitation disaggregation using the cascade model: 1h rainfall observed at Rio de Janeiro - São Cristovão on 05 Dec 2010 (blue). Based on statistical evaluations of long-term hourly precipitation series and their aggregation to coarser temporal resolutions, all relevant steps of the cascade disaggregation applied to daily totals are presented (green). The time series of each cascade level are shown for three realisations of the model $n = 1$ (left), $n = 2$ (centre), and $n = 3$ (right).





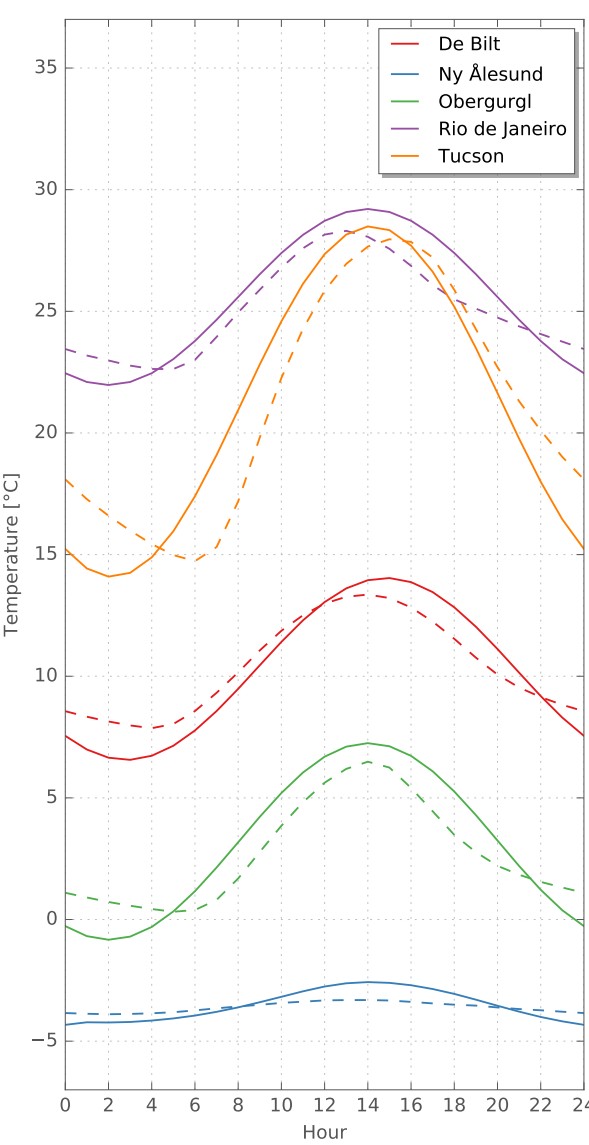

**Figure 8.** Long-term averages of diurnal courses of observed (dashed) and disaggregated (solid) temperature. Option T1 has been chosen and the sine-curve was modelled based on sunset and sun noon computations for minimum and maximum temperature, respectively. The period of time involved in this analysis is listed in Tab. 1 for each station.



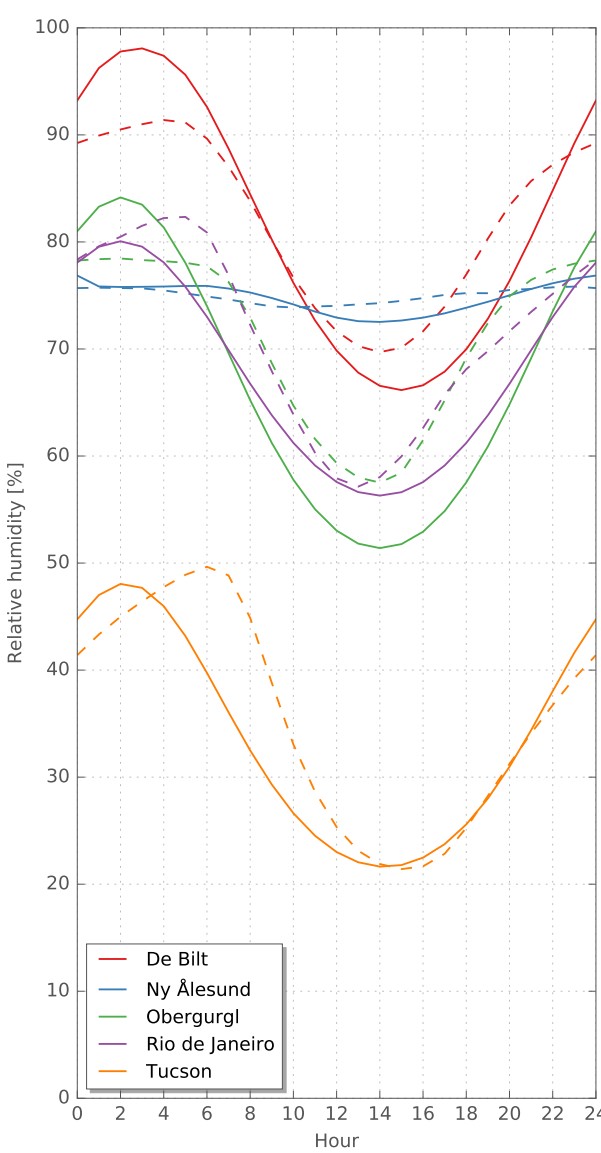

**Figure 9.** Long-term averages of diurnal courses of observed (dashed) and disaggregated (solid) relative humidity. The period of time involved in this analysis is listed in Tab. 1 for each station.





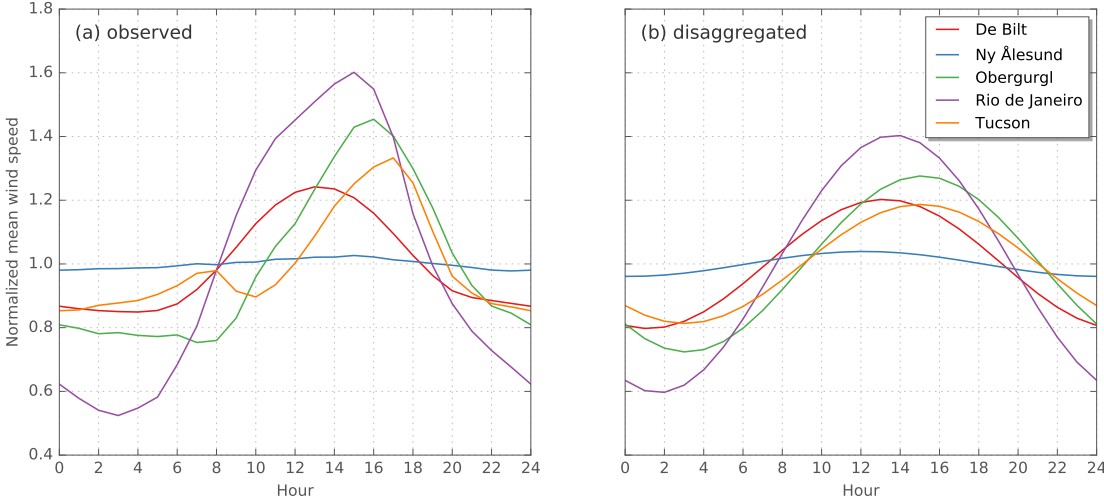

**Figure 10.** Long-term averages of diurnal courses of (a) observed and (b) disaggregated 'normative' wind speed. The 'normative' wind speed indicates the ratio of the long-term mean of the wind speed observed or modelled at a specified hour to the respective value averaged for the entire day. The period of time involved in this analysis is listed in Tab. 1 for each station.

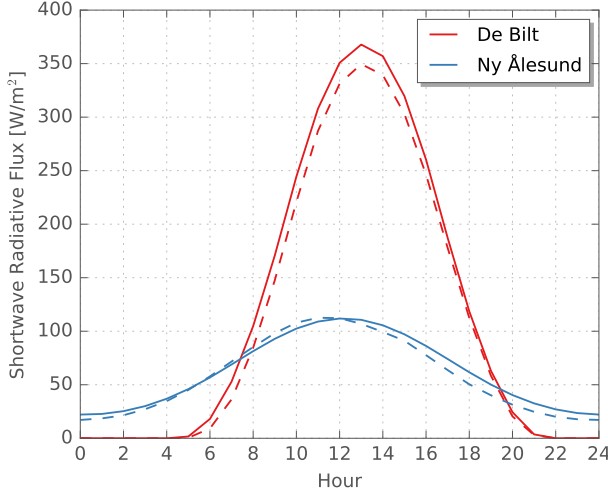

**Figure 11.** Long-term averages of diurnal courses of observed (dashed) and disaggregated (solid) shortwave radiation. Diaggregation is based on daily recordings of sunshine duration.




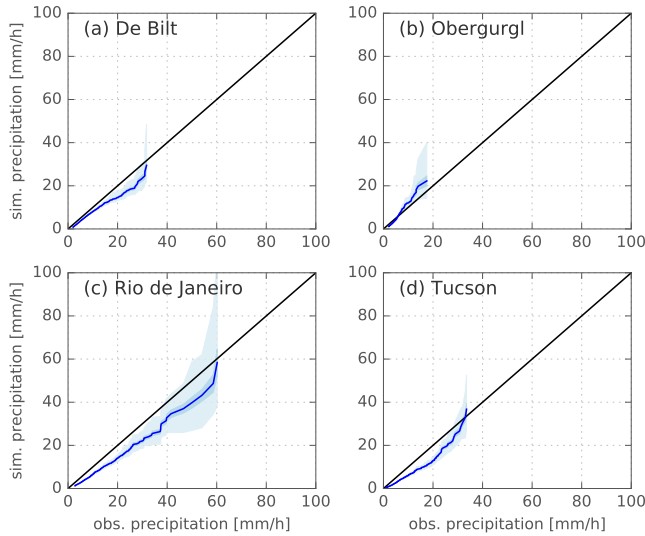

**Figure 12.** Modelled (blue) vs. observed (black) precipitation intensities for the 1% highest intensities derived using a split-sample test for the cascade model. For all panels the shaded areas refer to the standard deviation (dark blue) and range of values (light blue) computed for 100 realisations, respectively.

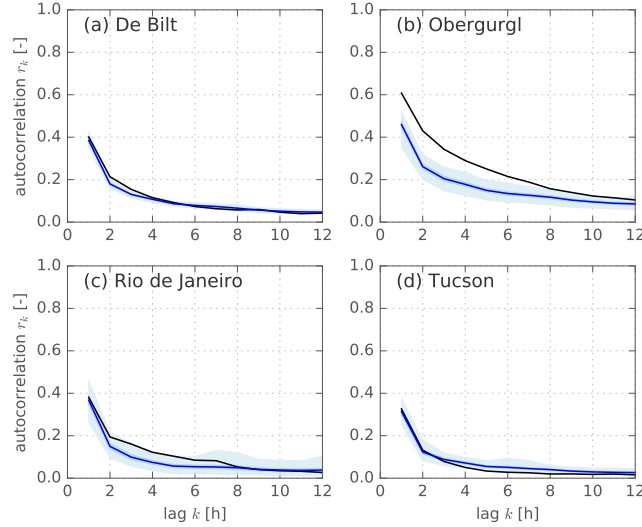

**Figure 13.** Autocorrelation $r_k$ as function of time lag $k$ (in hours) plotted for modelled (blue) and observed (black) precipitation time series. For all panels the shaded areas refer to the standard deviation (dark blue) and range of values (light blue) computed for 100 realisations, respectively.



**Table 1.** List of AWS investigated in this study. The elevation of each station $z$ is given in meters above sea level. Data availability refers to the available station recordings. The first period of time refers to the calibration period, whereas the second period is preserved for validation purposes. P=precipitation, T=air temperature, H=humidity, W=wind speed, R=solar radiation, S=sunshine duration. The location of each station is shown on the map in Fig. 2.

|   | Station | $z$ | Data availability | Source |
|---|---|---|---|---|
| 1 | De Bilt | 2 | 1961-1990, 1991-2014, P, T, W, H, R, S | KNMI (2015) |
| 2 | Ny-Ålesund | 11 | 1993-2004, 2006-2011, T, W, H, R, S | Maturilli et al. (2013a, b) |
| 3 | Obergurgl | 1938 | 2000-2007, 2008-2014, P, T, W, H, R | project data |
| 4 | Rio de Janeiro - São Cristovão | 5 | 2003-2008, 2009-2014, P, T, W, H | Alerta Rio (2015) |
| 5 | Tucson International Airport | 779 | 1973-1993, 1994-2014, P, T, W, H, R | NOAA (2015a); NREL (2015) |

**Table 2.** Overview of disaggregation methods included in *MELODIST*. The fist letter indicates the parameter that is considered by each method (P=precipitation, T=air temperature, H=humidity, W=wind speed, R=solar radiation, X=all variables). For each method, key references are given.

|   | Method | Type | Calib. |
|---|---|---|---|
| T1 | Standard Sine-redistribution with different options (Waichler and Wigmosta, 2003) | deterministic | no |
| H1 | $T_{dew} = T_{min}$ (Bregaglio et al., 2010; Waichler and Wigmosta, 2003) | deterministic | no |
| H2 | $T_{dew} = aT_{min} + b$ (Bregaglio et al., 2010; Waichler and Wigmosta, 2003) | deterministic | yes |
| H3 | Linear variation of $T_{dew}$ overlaid by sine function (Bregaglio et al., 2010) | deterministic | yes |
| H4 | $H_{min}$, $H_{max}$ (Bregaglio et al., 2010; Waichler and Wigmosta, 2003) | deterministic | no |
| W1 | Equal distribution | deterministic | no |
| W2 | Cosine function (Debele et al., 2007; Green and Kozek, 2003) | deterministic | yes |
| W3 | Random distribution (Debele et al., 2007) | stochastic | no |
| R1 | Scaling of potential shortwave radiation (Liston and Elder, 2006) | deterministic | no |
| R2 | Ångström (1924) model for sunshine duration S, then R1 | deterministic | yes |
| R3 | Bristow and Campbell (1984) model, then R1 | deterministic | yes |
| P1 | Equal distribution "$(\frac{1}{24})$" (Waichler and Wigmosta, 2003) | deterministic | no |
| P2 | Cascade model (Olsson, 1998) | stochastic | yes |
| P3 | Redistribution according to another station | deterministic | no |
| X1 | Linear interpolation | deterministic | no |



**Table 3.** Model performance measures for temperature disaggregation. $\bar{x}_o$ and $\bar{x}_s$ are the mean values of observed and disaggregated temperature, respectively. The standard deviation of the observed ($\tilde{\sigma}_o$) and disaggregated ($\tilde{\sigma}_s$) time series are also specified. The Root Mean Square Error (RMSE), the correlation coefficient $r$, and the Nash-Sutcliffe model efficiency (NSE) are calculated using the observed and disaggregated time series for each station.

| Station | $\bar{x}_o$ | $\bar{x}_s$ | $\tilde{\sigma}_o$ | $\tilde{\sigma}_s$ | RMSE | $r$ | NSE |
|---|---|---|---|---|---|---|---|
| | (unit of temperature [K]) | | | | | [-] | [-] |
| De Bilt | 283.57 | 283.45 | 6.88 | 6.94 | 1.74 | 0.97 | 0.94 |
| Ny Ålesund | 269.55 | 269.65 | 7.24 | 7.27 | 1.63 | 0.97 | 0.95 |
| Obergurgl | 275.90 | 276.36 | 7.87 | 8.03 | 2.00 | 0.97 | 0.94 |
| Rio de Janeiro | 298.25 | 298.74 | 4.03 | 4.34 | 1.66 | 0.93 | 0.83 |
| Tucson | 294.35 | 294.45 | 9.53 | 9.49 | 2.69 | 0.96 | 0.92 |

**Table 4.** Model performance measures for humidity disaggregation. $\bar{x}_o$ and $\bar{x}_s$ are the mean values of observed and disaggregated relative humidity, respectively. The standard deviation of the observed ($\tilde{\sigma}_o$) and disaggregated ($\tilde{\sigma}_s$) time series are also specified. The Root Mean Square Error (RMSE), the correlation coefficient $r$, and the Nash-Sutcliffe model efficiency (NSE) are calculated using the observed and disaggregated time series for each station.

| Station | $\bar{x}_o$ | $\bar{x}_s$ | $\tilde{\sigma}_o$ | $\tilde{\sigma}_s$ | RMSE | $r$ | NSE |
|---|---|---|---|---|---|---|---|
| | (unit of relative humidity [%]) | | | | | [-] | [-] |
| De Bilt | 81.82 | 81.63 | 15.12 | 15.48 | 12.67 | 0.66 | 0.30 |
| Ny Ålesund | 74.96 | 74.82 | 12.61 | 12.30 | 17.41 | 0.02 | -0.91 |
| Obergurgl | 70.83 | 66.42 | 17.67 | 13.40 | 16.43 | 0.51 | 0.14 |
| Rio de Janeiro | 70.95 | 67.45 | 14.13 | 10.76 | 10.39 | 0.72 | 0.46 |
| Tucson | 35.31 | 33.31 | 21.96 | 10.51 | 18.52 | 0.55 | 0.29 |

**Table 5.** Model performance measures for wind speed disaggregation. $\bar{x}_o$ and $\bar{x}_s$ are the mean values of observed and disaggregated wind speed, respectively. The standard deviation of the observed ($\tilde{\sigma}_o$) and disaggregated ($\tilde{\sigma}_s$) time series are also specified. The Root Mean Square Error (RMSE), the correlation coefficient $r$, and the Nash-Sutcliffe model efficiency (NSE) are calculated using the observed and disaggregated time series for each station.

| Station | $\bar{x}_o$ | $\bar{x}_s$ | $\tilde{\sigma}_o$ | $\tilde{\sigma}_s$ | RMSE | $r$ | NSE |
|---|---|---|---|---|---|---|---|
| | (unit of wind speed [m/s]) | | | | | [-] | [-] |
| De Bilt | 3.49 | 3.49 | 1.89 | 1.59 | 1.05 | 0.83 | 0.69 |
| Ny Ålesund | 4.03 | 4.03 | 3.23 | 2.57 | 1.95 | 0.80 | 0.64 |
| Obergurgl | 1.38 | 1.38 | 1.51 | 1.06 | 1.08 | 0.70 | 0.49 |
| Rio de Janeiro | 1.41 | 1.41 | 1.21 | 0.75 | 0.85 | 0.72 | 0.50 |
| Tucson | 3.27 | 3.27 | 2.07 | 1.17 | 1.70 | 0.57 | 0.32 |





**Table 6.** Model performance measures for radiation disaggregation. $\bar{x}_o$ and $\bar{x}_s$ denote the mean values of observed and disaggregated shortwave radiation, respectively. The standard deviation of the observed ($\tilde{\sigma}_o$) and disaggregated ($\tilde{\sigma}_s$) time series are also specified. The Root Mean Square Error (RMSE), the correlation coefficient $r$, and the Nash-Sutcliffe model efficiency (NSE) are calculated using the observed and disaggregated time series for each station.

| Station | $\bar{x}_o$ | $\bar{x}_s$ | $\tilde{\sigma}_o$ | $\tilde{\sigma}_s$ | RMSE | $r$ | NSE |
|---------|------|------|------|------|------|------|------|
| | (unit of radiative flux [W m$^{-2}$]) | | | | | [-] | [-] |
| De Bilt | 113.65 | 123.05 | 188.11 | 182.17 | 61.30 | 0.95 | 0.89 |
| Ny Ålesund | 59.73 | 63.21 | 92.42 | 89.38 | 31.90 | 0.94 | 0.88 |

**Table 7.** Model performance of the precipitation cascade model evaluated for each station. For the validation period, mean values are given for some relevant characteristics of the precipitation time series. An event is defined through consecutive hours with precipitation intensity greater than zero millimetres per hour. Numbers in brackets refer to the respective observed time series of each station.

| | De Bilt | Obergurgl | Rio de Janeiro | Tucson |
|---|---|---|---|---|
| Duration of events [h] | 3.91 | 4.72 | 3.41 | 2.90 |
| | (2.99) | (3.73) | (2.70) | (2.20) |
| Rainfall of events [mm] | 2.52 | 2.78 | 4.87 | 3.46 |
| | (2.45) | (2.78) | (4.63) | (3.81) |
| Duration of dry spells [h] | 21.76 | 23.05 | 39.24 | 118.44 |
| | (22.02) | (24.00) | (37.87) | (131.47) |
| Number of events per year | 342 | 316 | 206 | 72 |
| | (351) | (316) | (216) | (66) |