# Peer review of "An open-source MEteoroLOgical observation time series DISaggregation Tool (MELODIST v0.1.1)"

_Geoscientific Model Development, 2016_

## Referee Comment (RC1) · I. Pohle (Referee) · 3 May 2016

The manuscript by Förster et al. presents the software package MELODIST, a framework of state of the art methods for disaggregating meteorological time series. The methods included comprise deterministic and stochastic approaches with several options to choose for the individual meteorological variables. The disaggregation methods are described concisely with adequate reference to the relevant literature. The general applicability of the disaggregation methods is assessed by comparisons between observed hourly data and disaggregated hourly data based on daily variables. Therefore, five stations in contrasting climates have been chosen. The model code itself is well documented, the software package is easy to apply and modify and thus

has high potential on being used e.g. by hydrologist who require hourly input data for models. The manuscript is well structured and written. The methods are clearly documented and critically assessed both with reference to the literature and by own analyses of the authors. The conclusions are well supported by the results. I recommend the article for publication after minor revision for the following issues:

- Introduction: motivate the need of a disaggregation to hourly data more directly –for which purposes are data in hourly resolution needed (give examples)

- Introduction: while the relevant literature concerning disaggregation methods is addressed, reference to other tools / software packages for disaggregation of single meteorological variables (e.g. HyetosR) is missing

- Results: It is of interest, whether the distributions of the hourly data are preserved. Table 2 gives only mean values and standard deviations. Do the parameters of the distribution functions differ between observed and disaggregated hourly values?

- Results: On which basis have the times and locations for the result figures been chosen? Are these the times & locations where the disaggregation results fit the observations best? It might be helpful to add performance measures also for the time periods displayed.

Minor comments:

Page 1, line 1: Maybe specify: "Observations of hourly time series" / "Monitoring data in hourly resolution"

Page 3, line 24-26: Can be deleted, the reader should be familiar with the difference between deterministic and stochastic approaches.

Page 6, line 4: replace "small scale" with "sub-hourly"

Page 6, line 5: sentence unclear

Page 6 line 25: specify distribution (uniform)

General language comment: check when to use "a" and "an"

Page 10 line 18: why is this approach not referred to as "inverse distance weighting"

Page 13 line 2: replace "are not reproducable" with "is not reproducible" or: "cannot be reproduced"

Page 14 line 2 & line 29: these lines are redundant.

Page 15 line 3: can you give a ballpark figure on computational costs, e.g. disaggregation of 10 years of temperature data?

Page 15 line 5: give examples here (or in introduction)

Table 1: Please state whether "data availability" refers to hourly data

Figure 2: scale of the points – hard to perceive differences

---

## Referee Comment (RC2) · Anonymous Referee #2 · 23 May 2016

Review comments for gmd-2016-51

Overall, it is a well-done and useful paper. It brings together a number of methods together in one convenient place and software tool for the practitioner. When I next need to generate hourly meteorological data, I will refer to this paper and likely use the Python tool as well.

p. 2 In the discussion of the three approaches used for generating hourly timeseries, it would be helpful to point out even more explicitly to what degree each method can potentially reproduce the actual, time-specific values that represent actual history. The order of decreasing potential would be 1, 3, 2.

p. 6, line 14 Missing word "always" after "almost"

p. 6, Section 3.4.2 Suggest a stronger word than "overlie", perhaps "overpower" or "overwhelm or replace" to describe how a low-pressure system can be more important than local effects for wind generation.

p. 9, line 21 Misspelling: "releated"

p. 15, line 2 Would make more sense as "simple and easy-to-use"

Table 7 caption Suggest saying "parentheses" instead of "brackets"

p. 16, line 4 Instead of "warranted" a better phrase would be "inherent in the methodology".

---

## Author Comment (AC2) · 1 Jun 2016

**Reply to Anonymous Referee #2**

*(Reviewer's comments are in italics)*

**General comments:**

>   *Review comments for gmd-2016-51*
>
>   *Overall, it is a well-done and useful paper. It brings together a number of methods together in one convenient place and software tool for the practitioner. When I next need to generate hourly meteorological data, I will refer*

[Figure]

*to this paper and likely use the Python tool as well.*

We would like to thank Anonymous Referee 2 for for his/her positive evaluation of our manuscript and for his/her constructive comments and suggestions! The comments are highly appreciated and will help us to improve our manuscript. Please find below our detailed response.

**Specific comments:**

*p. 2 In the discussion of the three approaches used for generating hourly timeseries, it would be helpful to point out even more explicitly to what degree each method can potentially reproduce the actual, time-specific values that represent actual history. The order of decreasing potential would be 1, 3, 2*

Our first intention was to list these approaches according to their complexity in ascending order. We agree that sorting this list according to the potential regarding their capability of reconstructing the actual, time-specific values (i.e., the originally measured hourly values) would be beneficial. We will re-arrange the list accordingly. Moreover, we will take up your suggestion to point out more explicitly the potential of each method.

"In general, three completely different approaches exist (listed in descending order regarding their potential to reconstruct the originally measured hourly values that are representative for a given location and time):"

1. Disaggregation: "Despite their simplicity, disaggregation methods have great potential to reconstruct the originally measured hourly values for a given day as they are forced by actual daily values valid for that specific day."

2. Dynamical downscaling: "Since atmospheric (re-)analysis data represent the actual weather for a given time, dynamical downscaling of this kind of data is a sophisticated way to derive hourly values for that time and arbitrary locations in a realistic manner."

3. Weather generators: "[...] is different due to its random nature, which is why sub-daily time series do not provide the originally measured values."

Thank you for pointing us in this direction!

*p. 6, line 14 Missing word "always" after "almost"*

Done.

*p. 6, Section 3.4.2 Suggest a stronger word than "overlie", perhaps "overpower" or "overwhelm or replace" to describe how a low-pressure system can be more important than local effects for wind generation*

We replaced "overlie" by "obliterate". This term is used by Oke (1987) to describe this effect.

*p. 9, line 21 Misspelling: "releated"*

This typo is fixed.

*p. 15, line 2 Would make more sense as "simple and easy-to-use"*

Yes. We rewrote this wording accordingly.

*Table 7 caption Suggest saying "parentheses" instead of "brackets"*

Done.

*p. 16, line 4 Instead of "warranted" a better phrase would be "inherent in the methodology".*

You are right. Thanks!

**Literature**

Oke, T. R.: Boundary layer climates, Routledge, London, 2. edn., 1987.

---

## Author Response (AR1)

**Author's Response**

Title: An open-source MEteoroLOgical observation time series DISaggregation Tool (MELODIST v0.1.0)
Author(s): K. Förster et al.
MS No.: gmd-2016-51
MS Type: Model description paper
Iteration: Revised Submission

Dear Dr. Sander,

This document includes all modifications to the above mentioned manuscript. The referees' comments from the open discussion are listed below along with our detailed answers. Referee comments are in italics (blue). If required, we added some additional remarks (red) not listed in the open discussion so far.
In addition, we suggest some further changes, which are listed in a separate section of this document. Some changes have been personally communicated by Hannes Müller (University of Hannover) who made some remarks about the symbols of the precipitation cascade model. Moreover, he suggested adding some details regarding the cascade model along with two further references. We appreciate his hints which we would like to incorporate in the revised version of the manuscript if you accept these changes.

Thank you very much for handling our manuscript.

Best regards,

Kristian Förster
(on behalf of the authors)

**Table of Contents**

**Reply to review #1**

We would like to thank Dr. Ina Pohle for her detailed review of our manuscript and for her constructive comments and suggestions. The points raised in this review are highly appreciated and will help us to improve our manuscript. Please find our detailed response below.

**General comments:**

*The manuscript by Förster et al. presents the software package MELODIST, a frame-work of state of the art methods for disaggregating meteorological time series. The methods included comprise deterministic and stochastic approaches with several options to choose for the individual meteorological variables. The disaggregation methods are described concisely with adequate reference to the relevant literature. The general applicability of the disaggregation methods is assessed by comparisons between observed hourly data and disaggregated hourly data based on daily variables. Therefore, five stations in contrasting climates have been chosen. The model code itself is well documented, the software package is easy to apply and modify and thus has high potential on being used e.g. by hydrologist who require hourly input data for models. The manuscript is well structured and written. The methods are clearly documented and critically assessed both with reference to the literature and by own analyses of the authors. The conclusions are well supported by the results. I recommend the article for publication after minor revision for the following issues:*

*- Introduction: motivate the need of a disaggregation to hourly data more directly –for which purposes are data in hourly resolution needed (give examples)*

Based on available literature, we will add some examples and applications for which disaggregation methods are required. We will add one additional paragraph in the introduction:

"In contrast, hourly meteorological time series are required for numerous applications in geoscientific modelling. Typical applications in hydrology include both derived flood frequency analyses (e.g., Haberlandt and Radtke, 2014) and water balance simulations (Waichler and Wigmosta, 2003). In ecological modelling, sub-daily meteorological data are required for, e.g., the estimation of epidemic dynamics of plant fungal pathogens (Bregaglio et al., 2010)."

Thank you for pointing us in this direction.

*- Introduction: while the relevant literature concerning disaggregation methods is addressed, reference to other tools / software packages for disaggregation of single meteorological variables (e.g. HyetosR) is missing*

We will refer to the HyetosR package which we were not aware of. We appreciate this hint! The revised version of the manuscript will include a reference to this software (page two, first bullet point):

"For instance, the rainfall disaggregation package "HyetosR" (Kossieris et al, 2012, ITIA, 2016) provides an extensive parameter estimation methodology which is based on observed time series."

*- Results: It is of interest, whether the distributions of the hourly data are preserved. Table 2 gives only mean values and standard deviations. Do the parameters of the distribution functions differ between observed and disaggregated hourly values?*

We agree that the comparison of mean values and standard deviations only gives a simplified review of the distribution of these values. This is a valid point which we have discussed intensively. The variables addressed in the manuscript have different distributions which is why it is not possible to fit one single type of distribution function. For instance, temperature might be represented by a normal distribution for many sites, whereas precipitation is characterised by a lower limit of zero and asymmetry. To best possibly address the need for distributions and to keep the manuscript concise without extensive additions regarding theoretical distribution functions and parameter estimation, we decided to add an additional figure to the revised version of the manuscript including histograms of both observed and disaggregated values for each variable and each station.

A figure including histograms of both observed and disaggregated time series was added (Fig. 8 in the revised manuscript). Histograms are displayed for each variable and each station (if applicable). An additional explanation is now added to Sect. 4.1.:
"In order to gain some insight on how well the distributions of disaggregated time series match the observed ones, histograms for each variable and each site are displayed for both disaggregated and observed values in Fig. 8."

*- Results: On which basis have the times and locations for the result figures been chosen? Are these the times & locations where the disaggregation results fit the observations best? It might be helpful to add performance measures also for the time periods displayed.*

This question seems to refer to the example figures (Fig. 3 to Fig. 7) since only these figures include results of disaggregation methods for selected times and locations. In fact, the example figures for each variable have been randomly selected. They have been designed to show exemplarily how each of the methods work. You are right to say that this information needs to be clarified. In the revised version of the manuscript, we will explain in section 3.1 that the times and locations have been randomly selected. Adding performance measures for each method is a good point as this information would prove helpful. In principle, this is not a problem at all. However, this would require one additional table for each example plot. In our opinion, these additional tables would go beyond the scope of the exemplary type of figures. Therefore, we suggest adding the RMS error for each method to the legend in order to give an idea of model performance for each method for the times displayed in each figure (except for precipitation).

We have added the RMSE values for each method to the legend of Fig. 3, 4, 5, and 6, respectively. In section 3.1, we further state: "The subsequent sections provide details for each of the methods listed in Tab. 2. For each variable an example figure is provided which gives an idea of how each of the methods works. The times and locations of these figures have been randomly selected."

**Minor comments:**

*Page 1, line 1: Maybe specify: "Observations of hourly time series" / "Monitoring data in hourly resolution"*

We will rewrite this sentence accordingly: "Meteorological time series with one-hour time step are required in many applications in geoscientific modelling."

*Page 3, line 24-26: Can be deleted, the reader should be familiar with the difference between deterministic and stochastic approaches.*

We agree that most of the readers should be familiar with these terms. However, since the evaluation of stochastic methods requires multiple runs to perform statistical analyses, we believe that some introductory remarks might improve comprehensibility regarding the study design.

*Page 6, line 4: replace "small scale" with "sub-hourly"*

Done.

*Page 6, line 5: sentence unclear*

We agree that this sentence should be improved. We will revise this statement in the following way: "This idea best corresponds to averages of wind speed for a given increment of time (e.g., one hour) rather than instantaneous measurements."

*Page 6 line 25: specify distribution (uniform)*

This information was missing: "The function rnd is a random number generator which draws random numbers between 0 and 1 from a uniform distribution."

*General language comment: check when to use "a" and "an"*

We will review and correct the document with respect to the usage of "a" and "an". Thank you.

The manuscript has been updated with respect to this issue.

*Page 10 line 18: why is this approach not referred to as "inverse distance weighting"*

At present, this method simply transforms the mass curve from one station to another. Distance measures, which might be relevant if more than one highly resolved station is considered, are not considered in this method since the focus of the methods presented is on single sites only. However, a distance-related weighting considering more than one station can be easily applied to this method. This feature is implemented in the already cited IDWP program.

*Page 13 line 2: replace "are not reproducable" with "is not reproducible" or: "cannot be reproduced"*

Done.

*Page 14 line 2 & line 29: these lines are redundant.*

Yes. We removed the redundant sentence in line 29.

*Page 15 line 3: can you give a ballpark figure on computational costs, e.g. disaggregation of 10 years of temperature data?*

Thank you for this suggestion. The following information will be added to the revised version: "disaggregating 5 years of daily precipitation recordings using the cascade model takes less than 4 seconds on a notebook with a 2 GHz i7 CPU)"

*Page 15 line 5: give examples here (or in introduction)*

As pointed out earlier, we will add some examples in the introduction.

*Table 1: Please state whether "data availability" refers to hourly data*

Yes, "data availability" refers to hourly data. We will add this information to the caption.

*Figure 2: scale of the points – hard to perceive differences*

We slightly increased the dot size in order to improve perception. However, the difference between the two stations in Central Europe is small (De Bilt: 803 mm, Obergurgl: 851 mm).

Our first intention was to list these approaches according to their complexity in ascending order. We agree that sorting this list according to the potential regarding their capability of reconstructing the actual, time-specific values (i.e., the originally measured hourly values) would be beneficial. We will re-arrange the list accordingly. Moreover, we will take up your suggestion to point out more explicitly the potential of each method.

"In general, three completely different approaches exist (listed in descending order regarding their potential to reconstruct the originally measured hourly values that are representative for a given location and time):"

- Disaggregation: "Despite their simplicity, disaggregation methods have great potential to reconstruct the originally measured hourly values for a given day as they are forced by actual daily values valid for that specific day."
- Dynamical downscaling: "Since atmospheric (re-)analysis data represent the actual weather for a given time, dynamical downscaling of this kind of data is a sophisticated way to derive hourly values for that time and arbitrary locations in a realistic manner."
- Weather generators: "[...] is different due to its random nature, which is why sub-daily time series do not provide the originally measured values."

Thank you for pointing us in this direction!

*p. 6, line 14 Missing word "always" after "almost"*

Done.

*p. 6, Section 3.4.2 Suggest a stronger word than "overlie", perhaps "overpower" or "overwhelm or replace" to describe how a low-pressure system can be more important than local effects for wind generation*

We replaced "overlie" by "obliterate". This term is used by Oke (1987) to describe this effect.

*p. 9, line 21 Misspelling: "releated"*

This typo is fixed.

*p. 15, line 2 Would make more sense as "simple and easy-to-use"*

Yes. We rewrote this wording accordingly.

*Table 7 caption Suggest saying "parentheses" instead of "brackets"'*

Done.

*p. 16, line 4 Instead of "warranted" a better phrase would be "inherent in the methodology".*

You are right. Thanks!

**Literature**
Oke, T. R.: Boundary layer climates, Routledge, London, 2. edn., 1987.

**List of additional changes**

Page and line numbers refer to the marked-up version of the manuscript.

Page 1, line 22: To better acknowledge the ISD, we have added one additional reference (Smith et al., 2011).

Page 2, line 27: LAM denotes "Limited Area Model" instead of "Local Atmospheric Model". This term is updated now.

Page 8, line 13: We now state that the parameters a and b can be derived through optimization. This feature has been added to the software.

Page 9, line 6: We now state that the parameters A and C can be derived through optimization. This feature has been added to the software.

Page 9, line 15: We now specify that the cascade model proposed by Olsson is a microcanonical, multiplicative cascade model.

Page 9, line 22: "time step" is replaced by "temporal resolution"

Page 9, line 24: "equally spaced" is replaced by "equidistant"

Page 10, line 2: the term "branching generator" is added according to Olsson (1998)

Page 10, line 17: P(x/x) is replaced by P(x/(1-x)) since this notation is more exact, the same change was applied to w(x/x) which is now w(x/(x-1))

Page 11, line 4: "simply aggregated" is replaced by "transformed uniformly"

Page 11, line 6: the term microcanonical is specified here, too

Page 11, line 9: We have added a reference which describes the effect discussed in this paragraph, i.e. the lack of auto-correlation (Lombardo et al., 2012).

Page 11, line 15: In order to state more clearly the limitations of this approach, we have added some further remarks: "Areal peak intensities at sub-daily time steps might be overestimated due to this assumption which limits the universal applicability of this approach. However, this overestimation might be acceptable for some applications like, e.g., derived flood frequency analyses for hydrologic design purposes (Haberlandt and Radtke, 2014)."

Page 11, line 18: A newer reference describing an approach to handle spatial consistency is added (Müller and Haberlandt, 2016).

Page 11, line 27: We now state more clearly that this approach is restricted "to the period of time covered by recordings at one hour time step."

Page 11, line 31: We have recognized that the origin of daily data in the results section was not explained so far. Therefore, we have added this information as follows: "The time series used for disaggregation represent hourly observations aggregated to daily averages and totals, respectively."

Page 13, line 14: We now refer to the new Fig. 8 in Sect. 4.2: "This finding is also supported by the good agreement of the histograms constructed for both disaggregated and observed time series (Fig. 8, 1st column)."

Page 14, line 1: We now refer to the new Fig. 8 in Sect. 4.3: "Hence, minimum and maximum humidity are not preserved by this approach. This finding becomes apparent when considering the mismatch of minimum and maximum humidity reconstructions for some sites (e.g., Tucson, see Fig. 8, 2nd column for further details)."

Page 14, line 15: We now refer to the new Fig. 8 in Sect. 4.4: "This also becomes evident when observing the falling limb of the histograms of disaggregated values shown in Fig. 8 (3rd column)."

Page 15, line 1: We now refer to the new Fig. 8 in Sect. 4.5: "[…]and the coincidence of histograms computed for disaggregated and observed time series as displayed in (Fig. 8, 4th column)."

[revised manuscript text omitted]